# Identification of Molecular Markers Associated with Prostate Cancer Subtypes: An Integrative Bioinformatics Approach

**DOI:** 10.3390/biom14010087

**Published:** 2024-01-10

**Authors:** Ilaria Granata, Paola Barboro

**Affiliations:** 1High Performance Computing and Networking Institute (ICAR), National Council of Research (CNR), Via Pietro Castellino 111, 80131 Naples, Italy; 2Proteomic and Mass Spectrometry Unit, IRCCS Ospedale Policlinico San Martino, Largo R. Benzi 10, 16132 Genoa, Italy; pb300203@yahoo.it

**Keywords:** prostate cancer, castration-resistant prostate cancer, molecular profiling, data integration, precision medicine, essential genes

## Abstract

Prostate cancer (PCa) is characterised by androgen dependency. Unfortunately, under anti-androgen treatment pressure, castration-resistant prostate cancer (CRPC) emerges, characterised by heterogeneous cell populations that, over time, lead to the development of different androgen-dependent or -independent phenotypes. Despite important advances in therapeutic strategies, CRPC remains incurable. Context-specific essential genes represent valuable candidates for targeted anti-cancer therapies. Through the investigation of gene and protein annotations and the integration of published transcriptomic data, we identified two consensus lists to stratify PCa patients’ risk and discriminate CRPC phenotypes based on androgen receptor activity. ROC and Kaplan–Meier survival analyses were used for gene set validation in independent datasets. We further evaluated these genes for their association with cancer dependency. The deregulated expression of the PCa-related genes was associated with overall and disease-specific survival, metastasis and/or high recurrence risk, while the CRPC-related genes clearly discriminated between adeno and neuroendocrine phenotypes. Some of the genes showed context-specific essentiality. We further identified candidate drugs through a computational repositioning approach for targeting these genes and treating lethal variants of PCa. This work provides a proof-of-concept for the use of an integrative approach to identify candidate biomarkers involved in PCa progression and CRPC pathogenesis within the goal of precision medicine.

## 1. Introduction

In a recent demographic study, it has been observed that the increasing population ageing, due to demographic and social transitions, contributes to a rapid increase in new cancer cases (24 million in 2035), among which prostate cancer (PCa) still remains the leading cause of oncologic death in men worldwide [1]. As this pathology affects predominantly older men, cancer management can be complicated by comorbidity and age-related variations with significant social and economic implications. Considering the high impact that the burden of PCa has on families and health services, there is an urgent need to improve cancer surveillance and treatment, thus ensuring adequate disease management.

In the early stages of PCa development, tumour cell growth is dependent on circulating testosterone, providing the rationale for using androgen deprivation therapy (ADT) for localised disease. Under genetic and epigenetic alterations, PCa clones with a marked malignant phenotype evolve into the metastatic state (mPCa), which precedes the insurgence of castration resistance. The PCa clonal heterogeneity and gene instability, amplified by antineoplastic therapy effects, can be associated with the ineffectiveness of conventional ADT [2]. The early stages of “castration-resistant” prostate cancer (CRPC) are characterised by heterogeneous PCa cell populations generated under anti-androgens pressure, where the stress conditions allow adaptive cellular reprogramming to low circulating and tissue levels of testosterone. This condition leads to tumour progression and the proliferation of malignant clones through either androgen receptor (AR) pathway reactivation or AR-independent pathways activation [3,4]. As a consequence, in a single patient, it is possible to highlight extensive phenotypic heterogeneity characterised by the co-presence of cell populations with different evolutionary histories and drug susceptibility. Interestingly, it has been proposed that this diversity may originate from non-mutational mechanisms which result in an expansion of isogenic populations differing by their sensitivity to ADT [5]. Based on recent observations, it is hypothesised that ADT resistance results from the interplay between redundant genetic and epigenetic mechanisms engaged in complex crosstalk with cellular plasticity that facilitates adaptation to prolonged drug exposure [6,7,8].

Although multimodal approaches and new therapeutic resources improved metastatic CRPC (mCRPC) patients’ care, reliable prognostic and predictive criteria for selecting adequate therapy are missing, especially for high-risk patients. The lack of biomarkers able to guide clinicians to the most appropriate therapeutic choice led to weak prognosis improvement after the treatment of metastatic PCa (mPCa) and mCRPC [9]. The precision medicine inadequacy shown by several neoplasm treatments would depend on the absence of guidelines on target molecule selection based on clinical evidence, as defined by the ESMO Scale for Clinical Action of Molecular Targets—ESCAT [10]. In addition, although many studies have highlighted genetic alterations associated with PCa evolution, a significant role of epigenetic regulation has emerged in controlling the cancer cell plasticity involved in androgen-resistance acquisition [7,11,12]. The improvement of clinical practices, then, requires further studies to identify candidate target key genes and pathways associated with different PCa and CRPC cell populations.

The possibility of performing genome-wide gene deletion experiments, such as CRISPR-Cas9 and RNAi, allowed investigation of the gene essentiality in hundreds of cancer cell lines [13]. These studies highlighted the conditional nature of gene essentiality, a dynamic property that can change at the modification of genetic and/or environmental context. In particular, diverse sets of essential genes have been observed in different cancer tissues and even in cell lines deriving from the same tissue. These findings have great relevance in the view of precision cancer therapy. For instance, cancer-specific essential genes represent ideal candidates to target cancer cells while sparing healthy ones, in which those same genes are dispensable.

The increasing availability of public datasets and independent data derived from multiple studies allows the validation and generalisation of methods and findings, increasing their reliability. Many valuable studies providing useful insights and data are often treated as individual findings with the difficulty of deriving a unique reliable result that can be translated to clinical and applicative aims. Large-scale data integration from multiple experiments, although permitting an increase in the robustness of statistical tests by increasing the sample size, still presents challenging tasks related to the exact genetic and phenotypic correspondence of samples. Batch effect correction methods, which are often applied, risk overcorrecting the data, removing true but unknown biological differences [14]. To overcome these issues, an alternative approach can be the integration of the results obtained from the analysis of different datasets, searching for a consensus among the independent studies. 

Furthermore, while generally the investigation of essentiality starts from the definition of the context-specific essentialome to identify candidate biomarkers, here we present an inverse approach. We first identified gene sets relevant to PCa subtypes from a functional and clinical point of view, and then, we characterised the selected genes in terms of drug targeting and essentiality. 

The aim of this study was to identify and investigate key genes involved in PCa progression and CRPC evolution toward AR-dependent (AR+) or AR-indifferent (AR−) subtypes in order to derive candidate markers with functional driver roles useful in PCa patients’ clinical management. To this extent, we adopted a consensus-based approach for the analysis of multi-source data from reference works and the extraction of a concordant result, identifying two gene sets associated with differential expression patterns between clinically relevant classes: primary vs. metastatic PCa (PCa-gene set) and CRPC AR+ vs. CRPC AR- (CRPC-gene set). A growing body of knowledge highlighted that hitting multiple targets involved in cancer progression improves the therapy effectiveness [15,16,17]. Following this trend, our study aimed to select, among the compounds already in clinical use for other diseases, those targeting the multiple proteins coded by the PCa- and CRPC-gene set to facilitate the development of more effective therapies, especially for advanced stages.

## 2. Materials and Methods

### 2.1. Datasets

The present work was based on the integration of several published datasets in order to get a consensus list and a source of validation of genes with a crucial role in PCa progression and CRPC pathogenesis.

In particular, the comparison of metastatic versus primary PCa and mCRPC versus primary PCa have been performed using the following 8 datasets from Gene Expression Omnibus (GEO) portal: GSE3325 (expression profiling by array: Platform GLP570 Affymetrix Human Genome U133 Plus 2.0 Array) [18], GSE3933 (expression profiling by array: Platform GPL2695) [19], GSE68882 (GPL91 [HG_U95A] Affymetrix Human Genome U95A Array) [20], GSE32269 (GPL96 [HG-U133A] Affymetrix Human Genome U133A Array) [21], GSE6811 (GPL4747YN Human 36 K (sets 1–8)) [22], GSE70770 (GPL10558 Illumina HumanHT-12 V4.0 expression beadchip) [23], GSE6752 (GPL2891 GE Healthcare/Amersham Biosciences CodeLink™ UniSet Human 20 K I Bioarray) [24] and GSE35988 (GPL9075 Agilent-014698 Human Genome CGH Microarray 105A (G4412A)) [25]. 

The datasets published by studies in which gene lists involved in CRPC have been provided and here investigated and integrated with our findings are summarised in Appendix A and are GSE101607 (GPL10558 Illumina HumanHT-12 V4.0 expression beadchip) [26], GSE104786 (GPL5175 [HuEx-1_0-st] Affymetrix Human Exon 1.0 ST Array) [27] and GSE77930 (GPL15659 Agilent-016162 PEDB Whole Human Genome Microarray 4 × 44 K) [28] from GEO portal and SU2C/PCF (Dream Team, Cell 2015) study [29] from cBioPortal (https://www.cbioportal.org/, accessed on 10 November 2023).

Tissue-specific expression was independently validated in the following studies from cBioPortal: (i) PCa-gene set: Prostate Adenocarcinoma (PRAD-TCGA PanCancer Atlas) and Prostate Adenocarcinoma (MSK, Cancer Cell 2010) [30]; (ii) CRPC-gene set: SU2C/PCF (Dream Team, PNAS 2019) [31] and Neuroendocrine-PC (Multi-Institute, Nat Med 2016) [32].

### 2.2. Differential Expression Analysis

Differential expression analysis of the GEO datasets has been performed using the GEO2R tool of the GEO portal, which relies on the Limma R package [33], for the following comparisons:Metastatic versus primary PCa groups for GSE3325, GSE3933 and GSE68882 datasets;mCRPC versus PCa groups for GSE32269, GSE6811, GSE70770, GSE6752 and GSE35988 datasets;AR-driven versus non AR-driven groups for GSE101607;CHGA negative versus CHGA positive/SYP positive/SR negative groups were compared for GSE77930.

Genes with |log2fold-change| ≥ 1 and Benjamini–Hochberg adjusted *p*-value ≤ 0.05 have been considered significant.

A separate mention is needed for the SU2C/PCF (Dream Team, Cell 2015) dataset from cBioPortal. We selected the samples belonging to NE positive and negative groups based on the RPKM values of 3 well-known markers: *CHGA*, *SYP* and *AR*. We considered the mean value as the threshold and the values above and below it as positive and negative, respectively. This way, we obtained 7 *CHGA*-positive/*SYP*-positive/*AR*-negative and 29 *CHGA*-negative/*SYP*-negative/*AR*-positive samples. We downloaded the mRNA-Seq expression RPKM values and performed the “Group Comparison” analysis provided by cBioPortal, which, in case of continuous data of two groups, applies a one-sided *t*-test and adjusts the *p*-value using the FDR approach.

### 2.3. Over-Representation Analysis

Gene set enrichment analysis (GSEA) of the 32 genes (Appendix A) identified in our previous papers [6,34] was performed with the hallmark (H), chemical and genetic perturbations (CGP) and canonical pathway (CP) gene set collections from MSigDB (https://www.gsea-msigdb.org/gsea/msigdb/index.jsp, accessed on 10 May 2023). We considered the overlapping results with an FDR q-value < 0.05 to be statistically significant. After evaluating the top 100 results, we chose 43 relevant genes involved in the regulation of processes or pathways associated with PCa (Appendix A). Exploiting the GEPIA database (http://gepia.cancer-pku.cn, accessed on 10 May 2023), 18 genes were selected, performing ANOVA test to evaluate the differences between tumour and paired normal human prostate samples (|log2fold-change| ≥ 1, FDR-adjusted *p*-value ≤ 0.05) of the TCGA-Prostate Cancer dataset (PRAD) and their association with significant disease-free survival (log-rank < 0.05) (Appendix A).

Enrichment analysis of the gene sets was performed using the enrichR R package (v.3.2). Only terms having an FDR-adjusted *p*-value ≤ 0.05 were considered significant.

The correlation between drug sensitivity and gene expression was performed through the Gene Set Cancer Analysis platform (GSCA) (http://bioinfo.life.hust.edu.cn/GSCA, accessed on 26 June 2023) using data from the Genomics of Drug Sensitivity in Cancer (GDSC) database and Cancer Therapeutics Response Portal (CTRP).

### 2.4. Statistical Analysis

Principal component analysis was performed by PCAtools R package v.2.6.0 (https://github.com/kevinblighe/PCAtools, accessed on 10 November 2023) [35] using the following arguments: center = TRUE, scale = TRUE, removeVar = 0.1.

The Pearson correlation of samples was obtained using the corrr R package v.0.4.3 (https://CRAN.R-project.org/package=corrr, accessed on 15 October 2023) [36]. 

Receiver operating characteristics (ROC) curves were constructed using OriginPro 2021b software for statistical evaluation of the PCa- and CRPC-gene sets as potential molecular classifiers. The area under the ROC curve (ROC-AUC) was evaluated for each gene according to its deregulated expression (Appendix A) using test directions positive versus high (upregulated) and positive versus low (downregulated).

The Cox proportional hazards regression was applied to compute differential survival analysis on the gene expression data of PRAD PCa patients. The TPM (transcripts per million) normalized values from the Gene Data Common (GDC) portal were used. The cutoff value with the highest significance (lowest FDR) was determined, and in the case of equal values, the one with the highest hazard (HR) rate was selected. First, the univariate analysis was performed for single genes and then the mean expression of selected genes was calculated for analysing the survival probability in case of gene signature [37].

## 3. Results

### 3.1. Identification and Characterization of Clinically Relevant Gene Sets in PCa Progression to Metastatic and Castration-Resistant Phenotypes

In our recent papers, by using in vitro PCa models, we have identified 32 genes (Appendix A) associated with resistant phenotype acquisition after androgen deprivation treatments [6,34]. Here, additional computational analyses were conducted to assess the functional and clinical relevance of the 32-gene set and investigate the mechanisms involved in the PCa progression and acquisition of castration resistance (Figure 1).

To this extent, we performed a GSEA of the 32 genes overlapping with the hallmark (H), chemical and genetic perturbations (CGP) and canonical pathways (CP) gene set collections from MSigDB. We found 43 genes (Appendix A) significantly (FDR q-value < 0.05) associated with processes or pathways enriched by the 32 genes (see Section 2.3). Performing an investigation on the TCGA-Prostate Cancer dataset (PRAD) using the GEPIA database, we selected, among the above 43 genes, a subset of 18 genes (Appendix A) associated with significant disease-free survival and/or with differential expression levels in tumour vs. normal human prostate samples, thus extending our set from 32 to 50 genes (Appendix A).

Interestingly, by analysing data collected from the literature [34,38,39,40,41,42,43,44,45], among the 50 genes, 17 (34%) are associated with AR as coregulators of its transcriptional activity and/or defined as AR-controlled genes, and 24 (48%) are associated with prostatic neoplasms, castration resistance, invasiveness, metastasis or carcinogenesis according to the Gene Set to Diseases web platform (http://cbdm-01.zdv.uni-mainz.de/~jfontain/cms/?page_id=592, accessed on 15 October 2023) (Appendix A).

### 3.2. Evaluation of 50 Gene Set Alterations in PCa Tissue Samples at Different Evolutive Stages 

To identify a consensus among different experiments and to generalise our findings as much as possible, we performed differential expression analyses of the 50 gene set using mRNA abundances of patients showing the three different PCa stages from eight public datasets (Appendix A). Comparing mPCa to primary PCa, we found 27 (58%) differentially expressed genes (DEGs), including 15 up- and 12 downregulated genes (molecular profile 1, MP1). Two genes (*KLK3*, *QKI*) showed a discordant expression pattern and consequently were discarded. Similarly, 25 DEGs (50%) were obtained comparing mCRPC versus PCa, with 12 up- and 13 downregulated genes (molecular profile 2, MP2). One gene (*IL4*) was excluded due to a discordant expression profile.

Among the 34 genes included in MP1 and/or MP2 groups (hereinafter called PCa-gene set, Appendix A), 15 genes showed the same expression pattern (Figure 2A), and 16 were differentially expressed only in one of the two groups. Of note, an opposite expression profile for *ADAMTS1*, *ETV1* and *SMAD2* genes was observed between MP1 and MP2 (Figure 2A).

To investigate the potential application in clinical settings of the PCa-gene set, we evaluated their performance as molecular classifiers in PCa patients by performing a ROC curve analysis. To this extent, we exploited two multidimensional datasets containing both transcriptomics and clinical data: the PRAD-TCGA dataset with 493 samples of primary PCa and the Prostate Adenocarcinoma dataset (MSK, Cancer Cell 2010) including 181 primary PCa and 37 metastatic PCa samples. Significant AUC values (*p*-value ≤ 0.05) associated with disease-specific survival, progression, recurrence, metastasis presence and/or high recurrence risk group 1 [46] were shown for 20 genes (Figure 2B,C, Appendix A). Kaplan–Meier analysis and the log-rank test confirmed that low expression of *ADAMTS1*, *SPON2* and *EDN3*, evaluated individually, as well as a set, was associated with high-risk PCa-specific mortality (Figure 2D).

### 3.3. Identification of Genes Associated with Different CRPC Phenotypes 

To investigate the genes with key roles in CRPC, we exploited the findings of two other studies [26,29] that have previously identified genes associated with different CRPC phenotypes (Appendix A). The union of their genes with our PCa-gene set gave rise to a new list of 88 genes (Figure 3C).

The association between this new list and AR+ or AR− phenotype was investigated with data provided by Ylitalo et al. (GSE10167) [26], in which AR+ and AR− samples have been identified and labelled as AR-driven and non-AR-driven, respectively, with a strong unbalancing in favour of AR+ (32 AR+; 8 AR−).

From PCA representations, it is evident that these 88 genes show the same ability to discriminate the AR+ and AR− samples (Figure 3D) as the whole transcriptome (Figure 3A), confirming the selection of discriminating genes.

Furthermore, they seem to better explain the belonging of samples to AR+ or AR− groups, as shown by the correlation networks built using the gene expression data (Figure 3B,F). The networks have the nodes representing the samples and the edge width and length representing the correlation scores. This means that the more the samples are correlated, based on the mRNA abundance values, the closer they are together and the thicker the edges. In the case of using the expression values of the 88 genes, the AR− samples are mainly in the left part of the network, while all the AR+ samples are distant in the right one (Figure 3F). Instead, in the case of the whole transcriptome, the samples are all mixed up (Figure 3B).

The genes having the highest absolute loading scores in PC1, and thus giving the strongest contribution to the variance, were *STEAP2* and *CD40* (Figure 3E).

We further included two additional studies [27,28] in our analysis and searched for a consensus representative of CRPC AR+ or AR− phenotypes, where the latter is often associated with neuroendocrine (NE) features and identified as NE+, giving rise to three different profiles: adeno-CRPC (AR+/NE−), NE-CRPC with neuroendocrine features (AR−/NE+) or double-negative CRPC (DNPC) (AR−/NE−). 

To this extent, we extracted lists of differentially abundant genes from the four datasets listed in Appendix A as described in Section 2.

We first extended the 88 gene set by adding the marker genes identified by [28] (referred to as the Nelson list), obtaining a final list of 104 genes. These genes have been searched in the four differentially abundant gene lists to find a consensus. We considered concordant genes having the same expression pattern (up or down) and significantly different abundances between AR+ and AR− (FDR-adjusted *p*-value < 0.05) in at least two datasets. This way, we obtained a consensus list of 29 genes: *ALDH1A3*, *AR*, *CDKN2A*, *CHGA*, *CHGB*, *EHF*, *ENO2*, *EZH2*, *FKBP5*, *FOXA1*, *HES6*, *HNRNPK*, *HOXA13*, *HOXB13*, *KLK2*, *KLK3*, *NKX3-1*, *PCSK1*, *PIK3CB*, *PLPP1*, *PMEPA1*, *RB1*, *SCG3*, *SCN3A*, *SORD*, *SPON2*, *STEAP2*, *STEAP4*, *SYP* and *TMPRSS2*. In the following, this list will be referred to as the CRPC-gene set (Appendix A).

The CRPC-gene set showed its discriminative power on the AR+ and NE+ samples of the GSE77930 dataset (Figure 4), as well as on the AR+ and AR− samples of the GSE101607 dataset (Figure 5C,D), determining a clearer separation than using the Nelson list (Figure 5A,B).

### 3.4. Validation of CRPC-Gene Set on Independent Datasets

The CRPC-gene set was validated by verifying the association of different CRPC phenotypes with transcriptomic alterations using two independent datasets from cBioPortal: the SU2C/PCF (208 mCRPC samples/201 patients) and the Neuroendocrine-PC (49 samples/35 patients). For this purpose, mRNA expression of the CRPC-gene set was evaluated using ROC analysis.

In the SU2C/PCF study, tissue samples can be stratified into AR+ (AR-score ≥ 0.25), AR− (AR-score < 0.25), NE+ (NE-score ≥ 0.4) and NE- (NE-score < 0.4) [29]. The significant AUC values (asymptotic probability < 0.05) obtained for the differential expression level of the 30 CRPC genes (Appendix A, “Deregulated expression” column) indicated a clear discrimination of all four groups (Appendix A). Interestingly, using the NE-score cutoff of 0.4, the CRPC genes downregulated in AR− were able to classify with greater accuracy the NE+ samples. 

Considering the pathological classification available in the SU2C/PCF and the Neuroendocrine-PC studies, significant AUC values were obtained for all CRPC-genes. More specifically, in both examined studies, the genes with positive Log2FC (Appendix A) efficiently identified NE-CRPC samples, while the negative ones were associated with Adeno-CRPC samples (Appendix A; Figure 6A,B).

### 3.5. Functional Enrichment of Gene Sets

The functional role of the genes composing the three gene sets, as well as their co-involvement in processes and pathways, was assessed by performing the enrichment analysis of Gene Ontology biological process (GO-BP) and KEGG pathway terms and selecting the top 50 in terms of number of overlapping genes.

Most of the terms were shared between MP1 and MP2 genes, while different top-enriched terms were obtained using the CRPC set, confirming their important distinction (Figure 7). All sets enriched “Regulation of apoptotic process” from GO-BP and “Prostate cancer”, “Pathways in cancer” and “Pancreatic cancer” from KEGG. 

MP1 and MP2 genes were particularly involved in the regulation of transcription and processes related to tumoral transformation, such as angiogenesis, proliferation and differentiation. MP1 genes were more involved in biosynthetic processes, regulation of cell adhesion and response to stimuli, with MP2 genes instead in signalling pathways. CRPC genes enriched hormone-related pathways, inflammation response and membrane traffic, terms characterising the advanced status of the tumoral processes.

The KEGG pathways confirmed the shared and distinct roles highlighted from GO-BP terms, as MP1 and MP2 appeared to be involved in regulation and interaction processes—MP1 alone in adhesion molecules, cell cycle and proteoglycans, while MP2 in the acquisition of stem cells properties as an alteration of NOTCH signalling. CRPC genes enriched many cancer pathways. The significant results of the enrichment analysis are reported in Appendix A.

### 3.6. Investigating the Context-Specific Essentiality

The genes we identified as key players in the different phenotypes under study have been investigated in terms of context-specific essentiality.

To this extent, we exploited DepMap, the most curated portal containing data from experiments for the identification of essential genes in cancer [47].

We downloaded the gene effect scores (GEs) from CRISPR and RNAi gene deletion experiments on cell lines of prostate carcinoma. The more the score is negative, the more the gene is essential for the survival and growth of the cell. Considering the above results, we focused on *FOXA1*, *HOXB13*, *ETV1*, *ADMATS1*, *EZH2* and *AR* (Appendix A). All the genes showed context-specific essentiality as having a different behaviour in the different cell lines, remarking their potential as markers for targeted therapies. 

According to the CRISPR scores (shown in the left panel of the figure), all the genes showed essentiality in VCaP and/or LNCaP cell lines; particularly marked values were observed in the case of *FOXA1* and *HOXB13*. *FOXA1* showed highly negative values also in the case of 22Rv1 cells according to CRISPR and NCIH660 and VCaP according to RNAi (middle panel). *HOXB13* resulted strongly essential also in PC3 (RNAi). *ADAMTS1* showed slight essentiality for VCaP, PC3 and 22Rv1 cells in CRISPR experiments and in DU145 and VCaP in the case of RNAi. *EZH2* and *ETV1* also showed no strong essentiality, but the most negative values were obtained in LNCaP and MDAPCA2B cells for both genes and SHMAC5 only for *EZH2*. *AR* showed essentiality, particularly in LNCaP (RNAi) and 22Rv1 (CRISPR) cells, while it is not essential in the case of VCaP cells, that although highly express *AR* and are sensitive to the presence of androgens, do not need the *AR* signalling for proliferating [48].

It is worth noticing that an inverse proportionality can be observed between GEs and mRNA expression (right panel), as a confirmation that essential genes are strongly overexpressed in cancer.

### 3.7. Computational Drug Identification Based on the PCa- and CRPC-Gene Sets

In order to predict candidate compounds to treat aggressive subtypes of PCa, we explored the drug sensitivity of the PCa- and CRPC-gene sets using GSCA.

Correlating mRNA expression of the 20 genes upregulated in MP1 and/or in MP2 and drug sensitivity data from both GDSC and CTR resources, we highlighted 60 compounds with potential inhibitory activity on PCa progression (Appendix A).

Significant positive correlations were observed for *PPARG*, *ATP1B1*, *ASPH*, *ADAMTS1*, *VEGFA* and *NOTCH3* genes involved in different stages of PCa progression (Figure 8A). Interestingly, 17-AAG, a drug candidate in human clinical trials (phase 3), potentially exerted an inhibitory activity on *EZH2*, *PAX5* and *KHDRBS1* genes associated with the risk of developing metastatic phenotypes (Figure 8A).

Similarly, performing the analysis for the CRPC-gene set (Appendix A), we identified 60 candidate chemicals to be used for Adeno-CRPC in combination with or alternatively to anti-androgen therapy (Appendix A) and 25 for NE-CRPC treatment (Appendix A). The overexpressed genes *STEAP2*, *PMEPA1*, *PLPP1*, *ALDH1A3*, *FOXA1* and *EHF* likely are drug targets in the case of Adeno-CRPC treatment, while *ENO2*, *HES6* and *EZH2* for treating the NE-CRPC (Figure 8B).

It is worth noticing a significative relationship between the expression of genes found to be essential in the previous analysis (*FOXA1*, *HOXB13*, *ETV1*, *ADMATS1*, *EZH2* and *AR*) and 60 chemicals, among which several were already identified in the above analyses (Appendix A).

We selected, among the drugs shown in Appendix A, the 20 FDA-approved drugs for the treatment of different cancer histotypes and potentially to be evaluated also for the advanced forms of PCa (Appendix A). Of note, 14 compounds have already been tested in preclinical and clinical studies on their potential efficacy for PCa and CRPC treatment, and 11 were potentially active on essential genes. Drugs that exerted an inhibitory effect on epigenetic regulation of transcription seem to have increased activity in androgen-dependent forms, while those which interfere with MAPK-regulated signal transduction pathways could be more effective on the NE-CRPC.

We evaluated the interactions between the upregulated genes in PCa, CRPC AR+ or CRPC AR− (Appendix A) and the drug targets associated with epigenetic transcription regulation or MAP kinase signal transduction regulation (Appendix A) [8,49,50,51,52,53,54,55,56,57,58,59,60,61,62,63,64,65,66,67,68,69,70,71]. Overall, they are involved in multiple significant connections supported by low enrichment *p*-values, which indicate that these proteins have more interactions than expected and are, thus, biologically connected (Appendix A). This result follows the therapeutic trend of selecting compounds that, by targeting central proteins involved in many physical and functional connections, may have an impact on multiple pathways [72].

Interestingly, using this approach, we observed that 17-AAG (Appendix A), a HSP90 inhibitor, exerts the inhibitory effect on its specific target and likely also on EZH2, PAX5, NOTCH4 and KHDRBS1. Indeed, the PPI network shows a strong physical and functional connection between the proteins coded by the upregulated genes of the PCa-gene set (Appendix A) and HSP90 (enrichment *p*-value 1.8 × 10^−4^), mainly due to a co-involvement in transcription-related pathways (enrichment tables in Appendix A).

## 4. Discussion

The significant efforts made to study the molecular mechanisms involved in PCa progression identified molecular alterations potentially relevant for improving the clinical management of patients [73,74]. Over the past decade, the use of new drugs inhibiting the *AR* axis improved the treatment of mPCa and some forms of CPRC, but these chemicals have led to increased CRPC with neuroendocrine features that are still incurable [3,51]. Hence, the hypothesis that PCa progression may depend not only on genetic alterations but also on epi- and/or non-genetic factors caused by drug-stress pressure. For this reason, researchers are currently focusing on identifying the key mechanisms and the master genes driving PCa cells’ fate toward metastatic and resistant phenotypes to develop novel and more effective therapeutic approaches [7,8,9,11,75].

Heterogeneity is an intrinsic characteristic of PCa that accentuates during neoplastic evolution, particularly in the advanced phases of disease and in the acquisition of resistance to anti-androgen treatments. Over the years, numerous in vitro and ex vivo models have been developed [76] in an attempt to mimic human PCa evolution and study the biological mechanisms and the key proteins involved in cell growth and proliferation leading to neoplastic progression. Studies performed using preclinical models have identified molecular alterations as potential markers for a specific pathological stage and/or as therapeutic targets [10]. Nonetheless, due to the intrinsically static and homogeneity of the in vitro models, frequently, the results obtained were not suitable to be translated into clinical practice.

Starting from 32 proteins previously identified in in vitro models mimicking the early stages of androgen-resistance [6,34] and using an integrative bioinformatics approach (schematically illustrated in Figure 1), we derived two gene panels that dynamically track the evolution of phenotypic changes: the PCa-gene set and the CRPC-gene set (Appendix A). The 34 genes included in the PCa-gene set, according to their expression in PCa tissues, were divided into two groups: (i) MP1, containing genes altered during primary PCa progression toward metastatic phenotype and (ii) MP2, involving genes modified by anti-androgen resistance acquisition (Figure 2A). The CRPC-gene set, instead, was based on the expression profile of 30 genes discriminating between the AR-driven (Adeno-CRPC) and the AR-indifferent (NE-CRPC) resistant phenotypes. To clarify the role of the PCa- and CRPC-gene set in prostate carcinogenesis, we have schematically depicted their changes in gene expression (Appendix A), diagnostic efficiency (Figure 2 and Figure 6) and drugability (Appendix A) in Figure 8.

Nine genes (*AR*, *EZH2*, *FOXA1*, *HOXB13*, *HOXA13*, *KLK3*, *EHF*, *SORD* and *SPON2*) were shared by both PCa- and CRPC-gene sets suggesting their potential role in the multiple evolutive phases of PCa progression (Figure 8). The percentage of genes participating directly or indirectly in the *AR* activity increased from 47% for those shared between MP1 and MP2 (7 over 15 genes) to 78% (7 over 9 genes) for genes common to PCa- and CRPC-gene lists. These results confirmed the essential role of *AR* in regulating PCa cells growth and proliferation during the early stages of the oncogenesis, as well as in the resistance acquisition [3,74]. It has been reported that the anti-androgen therapy pressure exerted on the intrinsic cellular heterogeneity of PCa tissues led to the selection of tumour cells showing *AR* gene alterations or epigenetic perturbation of its regulated pathways [8,11]. Consequently, during the tumour progression, two different phenotypes can be developed: androgen-responsive, where *AR* always acts as a driver gene, or androgen-insensitive, independent of *AR* activity [75]. In agreement with this statement, we observed that *AR* was highly expressed in the stages leading to mPCa and Adeno-CRPC, while it was poorly expressed in NE-CRPC (Figure 8). 

The ROC curve analysis of multidimensional datasets including transcriptomic and clinical data of PCa tissues from the Prostate Adenocarcinoma study (MSK, Cancer Cell 2010) revealed that the transition to the metastatic phenotype was significantly associated with overexpression of *AR*, *EZH2*, *NOTCH3*, *ZIC2* and *SOX4* genes and downregulation of *ALDH3A2*, *EHF*, *EPHA3*, *KLK3*, *SORD*, *XBP1*, *ADAMTS1* and *PRDM5* genes (Figure 2B). The correlation between NOTCH3-MMP3 axis activation and bone metastases induction [77], the pro-metastatic activity of EHF knockdown in PCa cells [78] and the involvement of *SOX4* and *EZH2* in enhancing the PCa cells invasiveness toward the activation of AKT and β-catenin pathways [11] have been experimentally demonstrated. According to the above, high levels of *SOX4* and *EZH2* were observed in the patients included in the high recurrence group 1 of the TGCA-PRAD study (Figure 2C).

The PCa progression toward the androgen-refractory state was exclusively correlated, in agreement with previous studies [79,80,81], with low expression levels of *ALDH3A2* and increased expression of *NOTCH1*, *NOTCH4* and *QKI* (Figure 8), that significantly identified patients with poor prognosis (Figure 2B,C). 

Our investigation interestingly highlighted that ADAMTS1 may act, in the context of PCa tissue, as a tumour suppressor or as a pro-tumorigenic factor in agreement with previous experimental evidence [82]. Furthermore, the downregulation of *ADAMTS1* was associated with the enhancement of tumours toward metastasis, while high levels were observed in CRPC. Recently, it has been reported that, depending on tumoral contexts, ADAMTS1 can induce proteolytic extracellular matrix modification, activating cell plasticity, a biological process involved in the acquisition of therapeutic resistance [83]. It is not surprising that the low level of *ADAMTS1* in PCa tissues had a significant prognostic value as a disease-specific marker (Figure 2D). 

Using two datasets reporting tissue-specific transcriptomic data and histopathologic information of CRPC patients, we verified the high discrimination power of the CRPC-gene set in identifying the expression patterns associated with Adeno- or NE-phenotype (Figure 6). Six genes associated with AR activity (*AR*, *TMPRSS2*, *HOXB13*, *NKX3-1*, *FKBP5*, *ALDH1A3* and *PMEPA1*) and *PLPP1*, a gene involved in sphingolipid metabolism and linked to androgen signalling [84], were overexpressed in Adeno-CRPC tissues of both datasets (Figure 6), in agreement with the described role of *AR* in these patients [3]. In NE-CRPC samples, the six molecular markers of NE phenotype *CHGA*, *CHGB*, *ENO2*, *PCSK1*, *SCG3* and *SCN3A* [73,85,86] were upregulated, as well as *HES6* and *EZH2* genes, involved in the regulation of cell fate decision. Ramos-Montoya et al. [87] reported that *HES6* had a driving role in androgen-independence acquisition by activating AR-independent pathways for sustaining the survival of PCa cells treated with anti-androgen therapies. Interestingly, *EZH2* and *CDKN2A* were included in the HES6-associated gene signature and were strongly connected with unfavourable outcomes in PCa patients. In our study, with respect to genes involved in *AR* signalling, an inverse behaviour was observed for the expression of EZH2. Indeed, it gradually increased as the tumour progressed toward metastatic and castration-resistant forms (Figure 8), in agreement with experimental evidence previously reported [11,88]. It has been suggested that EZH2, acting as an epigenetic regulator, controls cell cycle and cell stemness and promotes epithelial–mesenchymal transition (EMT), metastatic progression in PCa and neuroendocrine transdifferentiation in CRPC [12,89,90]. 

Our work, in agreement with recent experimental evidence, highlighted that, in addition to the above-mentioned genes, *FOXA1*, *HOXB13* and *ETV1* have a pivotal role in supporting cell growth and proliferation during the stages of PCa evolution (Figure 8). FOXA1 and HOXB13, acting as pioneer factors (PFs) found in chromatin remodeler complexes, reprogram *AR* transcriptional activity [91]. In normal prostate cells, these PFs regulate *AR* accessibility to activate the transcriptional processes while, in tumoral cells, their altered availability triggers aberrant transcription programs leading to the different oncogenic phenotypes of the evolutive forms of PCa [11,90,91]. Consequently, in accordance with the AR behaviour, the expression of *FOXA1* and *HOXB13* genes was inversely correlated with the aggressiveness of PCa forms such as mPCa, mCRPC and NE-CRPC (Figure 8). 

The ETS transcription factor ETV1 cooperates with *AR* in regulating the transcription of androgen-driven genes involved in cell growth and proliferation. Baena et al. [92] have found that ETV1 overexpression activates, in concert with AR, an oncogenic program in PCa cells leading to metastatic phenotype and a worse patient outcome. Although, in our analysis, the *ETV1* expression was upregulated in localised vs. metastatic PCa, a lower expression level of this gene was detected in mCRPC with respect to PCa (Figure 8). The role of *ETV1* in CRPC subtypes is still unclear, although it has been observed that the interplay between TGF-β/SMAD signalling and the ETV1 oncogenic activity could be regulated depending on the cellular genetic context [93]. We hypothesised that SMAD2, an intracellular signal transducer of TGF-β, is likely to be involved in this mutual interaction, as we observed a concordant expression pattern for *SMAD2* and *ETV1*, which were overexpressed in mPCa and downexpressed in mCRPC (Figure 8). In support of this hypothesis, it has been pointed out that, in the early stage of PCa development, the TGF-β signalling maintains cell homeostasis by acting as a tumour suppressor, while, in the advanced stages of the disease, it promotes cell proliferation and dedifferentiation [94]. 

The enrichment analysis performed on MP1 and MP2 gene sets evidenced common processes and pathways associated with the dynamic switching of PCa cells between proliferative, metastatic and resistant phenotypes such as the regulation of signal transduction, transcription, differentiation and proliferation (Figure 7 and Appendix A). This analysis also revealed other biological processes and pathways promoting cell plasticity that, under androgen deprivation therapies, can facilitate the acquisition of stem cell properties. In this regard, MP2 genes were associated with NOTCH, TGF-β and BMP pathways that controlled cell proliferation and differentiation in CRPC, unlike the normal prostate tissue in which they maintain homeostasis [3,13,66,95]. 

As expected, among the CRPC genes, the ones upregulated in Adeno- vs. NE-CRPC contributed to androgen resistance toward the aberrant AR signalling pathway, while neuroendocrine transdifferentiation was sustained by increased *EZH2* and *HES6* expression, promoting androgen independence and the neuroendocrine phenotype acquisition of PCa cells [2,88,90]. 

From the previously reported data, both gene sets were involved in sustaining the dynamic evolution of PCa and could be considered as potential targets to develop new therapeutic strategies aimed at intercepting the different stages of PCa. The activation of bypass pathways and the complex nature of signalling networks that involve feedback loops and crosstalk are responsible for the poor effectiveness of monotherapy. The assumption that the therapy effectiveness and some observed drugs’ contradictory effects are tumour-dependent is another issue to be considered in designing new therapeutic strategies. Consequently, the identification of clinically relevant predictive biomarkers is necessary to individuate specific and effective treatments. On the assumption that a single drug targeting multiple pathways can improve therapeutic efficiency [15,16,17], using the GSCA platform, we extracted the chemicals having a positive and significant correlation with the expression of PCa- and CRPC-gene sets. Globally, our results showed that inhibitors of regulators controlling epigenetic transcription and MAPK signal transduction (Appendix A) can represent promising and alternative drugs to counteract PCa progression and resistance acquisition. The findings of our study, in agreement with other experimental evidence [75,88,92], provide the rationale for testing the efficacy of epigenetic drugs in the treatment of PCa and Adeno-CRPC, as the molecular alterations observed in PCa evolution toward metastatic and resistant phenotypes mainly affect transcription factors and coregulators involved in AR activity (Figure 8, Appendix A). It has also been reported that the histone deacetylase inhibitors panobinostat and vorinostat simultaneously block the *AR* expression and inhibit the transcription of genes under their control [51]. Conversely, treatment with molecules targeting proteins regulating BRAF and MEK signal cascade and the TGF-β pathway can be used for NE-CRPC, suggesting that these pathways had a pivotal role in sustaining the neuroendocrine phenotype. In this regard, Vellano et al. [13] observed that the treatment of melanoma with BRAF/MEK inhibitor improved recurrence-free survival more efficiently in female than in male patients and, using preclinical models, attributed this significant difference to *AR* inactivity. This intriguing result had a relevant clinical implication of considering drugs such as dabrafenib, dasatinib, selumetinib and trametinib for the treatment of CRPC as monotherapy or in combination with ADT [8,66,67]. A recent study provided evidence for repositioning the antifungal ciclopirox (CPX), an iron chelating compound, for cancer therapy [71]. CPX exerted its anti-tumoral effect, inducing apoptosis and inhibiting cell proliferation, migration and angiogenesis. The PCa and CRPC cells treatment with CPX impaired the (WNT)/β-catenin pathway that, in advanced tumoral forms, regulated cell plasticity, inducing cancer cell stemness [75]. Therefore, it is not surprising that the “iron ion transport” was included in the enriched GO-BP terms of the CRPC-gene set (Figure 7) and the CPX was listed among the drugs to be considered for PCa and Adeno-CRPC (Appendix A).

The FDA has not approved any HSP90 inhibitors so far, but the clinical potential of *HSP90* inhibition for treating tumours should be taken into account for mPCa therapy. In this study, 17-AAG (Appendix A), an HSP90 inhibitor candidate in human clinical trials (multiple myeloma; phase 3), potentially exerted an inhibitory activity on *EZH2*, *PAX5* and *KHDRBS1* genes associated with the risk of developing metastatic phenotypes (Figure 8) by modulating gene expression (Appendix A).

Taken together, these results suggested that there is a set of genes (*AR*, *EZH2*, *FOXA1*, *HOXB13*, *HOXA13*, *KLK3*, *EHF*, *SORD*, *SPON2*, *ADAMTS1*, *ETV1* and *SMAD2*) involved in modulating different transcriptional programs that may determine the phenotype-specific expression profile during PCa progression. It should be noted that among those, six genes showed context-specific essentiality, involved in sustaining PCa cell growth and differentiation (Figure 8). Although what is observable in vitro could not be exactly reproduced in an alive organism, in vitro experiments are still considered an important and valuable strategy to get useful insights, and the variety of the genetic background of the different cell lines gives the possibility to evaluate the biological behaviour in different scenarios. Accordingly, these results represent an additional clue for confirming the contribution of the selected genes in discriminating the phenotypes under investigation. The therapeutic implication of these findings was clear by analysing data from the GSCA platform (Appendix A), according to which 55% of drugs can inhibit these essential genes. 

He et al., in an exhaustive review [52], reported the recent therapeutic advances in improving the clinical management of PCa patients, describing several chemicals targeting cell signalling or pathways associated with specific molecular alterations, in addition to AR-signalling inhibitors, representing the gold standard for the treatment of androgen-sensitive PCa forms. Interestingly, among the new drugs proposed by authors, panobinostat, vorinostat, dasatinib and temsirolimus are included in Appendix A of the present study. The other molecules we have identified can likely provide further helpful knowledge for designing or repositioning drugs, thus contributing to achieving the goal of precision medicine in PCa treatment.

## 5. Conclusions

This study provided a proof-of-concept for demonstrating the benefits of using an integrative bioinformatic approach, that, by joining the recent literature and data, was able to highlight the master genes involved in non-mutational mechanisms essential to support cell growth and differentiation toward metastatic and androgen-independent PCa phenotypes. In our opinion, these findings may provide useful insights to develop therapeutic strategies supporting personalised medicine. We are aware that in some cases the methodological approaches adopted in the current work can present some limitations. This was essentially due to the availability of published data, in terms of both their nature and processing level. We think that the consensus strategy, according to which the results must be confirmed by at least one more dataset to be taken into account, allowed us to derive interesting insights despite the limitations, since each analysis represented an additional clue to increase the reliability of our findings.

## Figures and Tables

**Figure 1 biomolecules-14-00087-f001:**
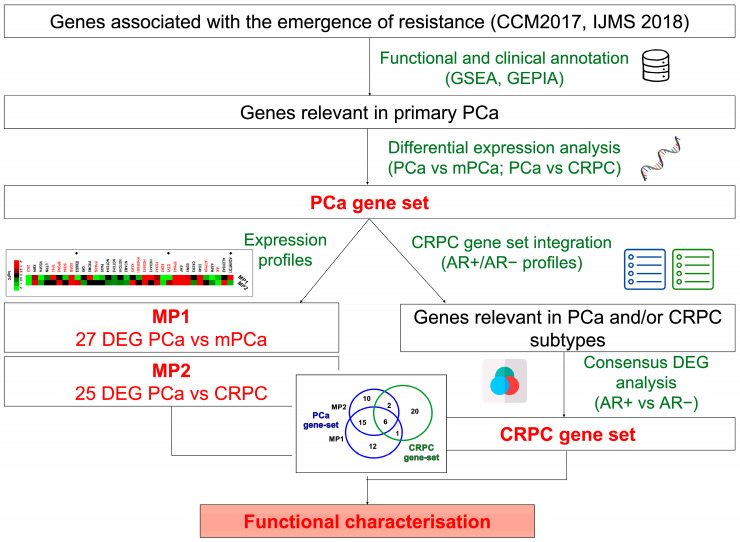
A schematic illustration of the workflow adopted in the current study. In the boxes are the gene sets that represent the input and output of the investigation processes, while out of the boxes, in green, are the analyses that allow the definition of the gene sets. The ultimate action in the filled box at the bottom of the figure was the functional characterization of the gene sets identified.

**Figure 2 biomolecules-14-00087-f002:**
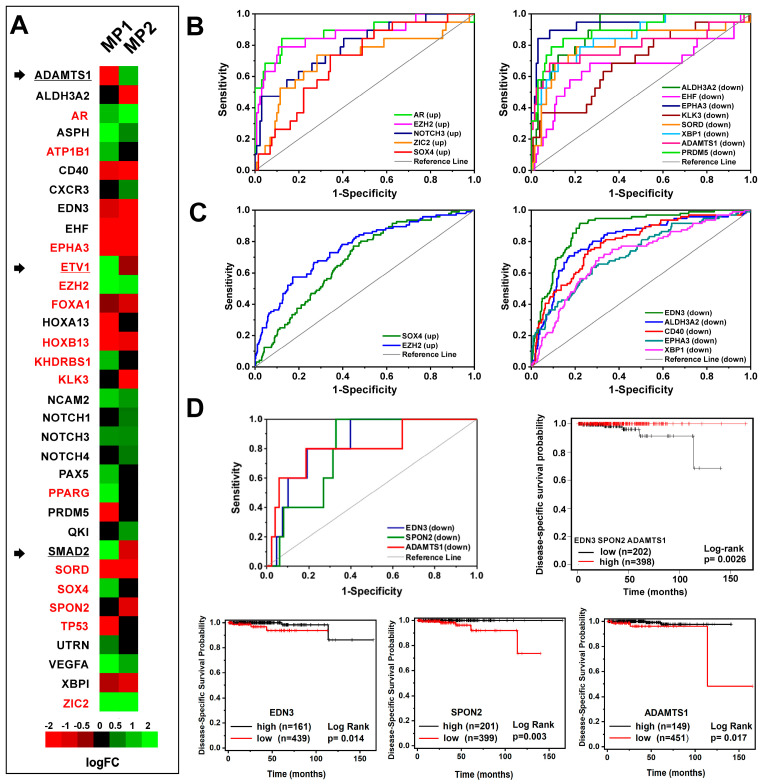
Evaluation of the PCa-gene set expression in PCa tissues at different tumour grades. (**A**) Heatmap showing the differential expression of MP1 (mPCa vs. PCa) and MP2 (mCRPC vs. PCa) genes. AR-associated genes are coloured in red. Arrows indicate genes with opposite expression in MP1 and MP2. The ROC curve analysis of PCa-gene set demonstrates its potential value as a molecular classifier to discriminate between patients with vs. without metastasis by MSK, Cancer Cell 2010 (150 patients) (**B**) and included vs. not included in high-recurrence risk subgroup 1 by PRAD-TCGA, PanCancer Atlas study (493 patients) (**C**). (**D**) ROC curve analysis and Kaplan–Meier plots showing disease-specific survival of PCa patients (PRAD-TCGA, PanCancer Atlas study) with high or low expression level of *ADAMTS1*, *SPON2* and *EDN3,* both individually and as a set. For ROC curves, only significant AUC values > 0.6 (asymptotic probability < 0.05) are reported.

**Figure 3 biomolecules-14-00087-f003:**
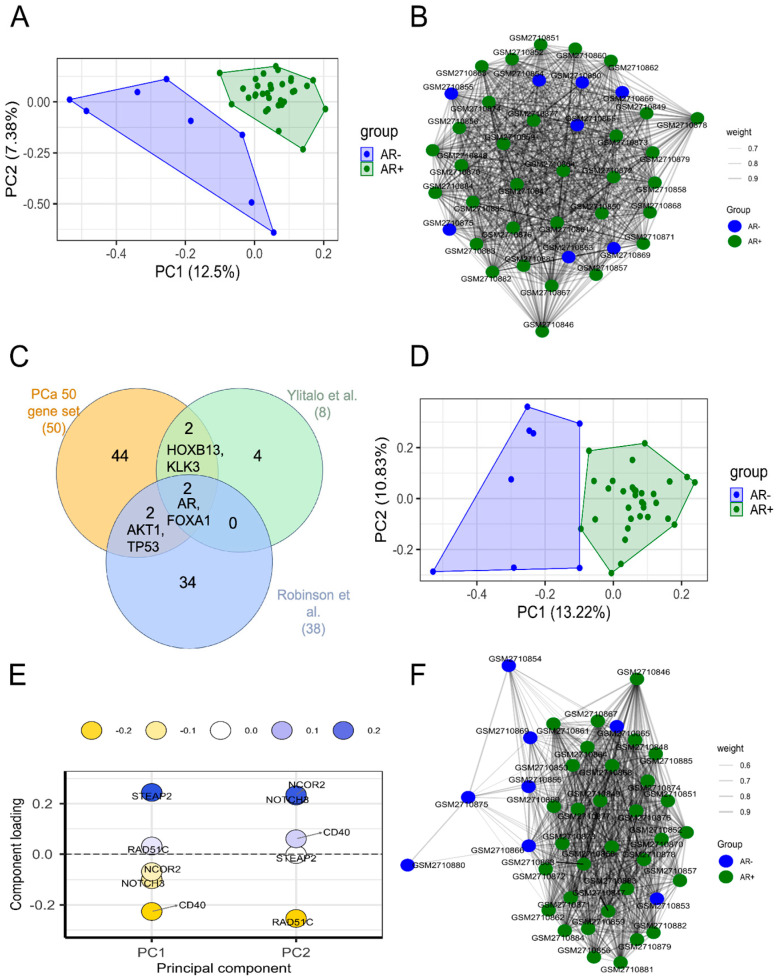
(**A**,**B**) PCA and correlation plots of AR+ (green) and AR− (blue) samples using abundance values of all genes from the dataset GSE10167. PC1 and PC2 variation percentages are also shown. In the correlation plot, the samples are the nodes labelled by sample ID and the edge thickness is proportional to the correlation score. (**C**) Venn diagram showing intersection among PCa-gene set containing 50 genes (orange circle), 8 genes identified by Ylitalo et al. [26] (green circle) and 38 genes identified by Robinson et al. [29] (blue circle). The overlapping genes are shown; (**D**) PCA plot of AR+ (green) and AR− (blue) samples using abundance values from the dataset GSE10167 of 88 genes obtained summing up the gene lists from PCa-gene set [26,29]. (**E**) Loadings plot of PC1 and PC2 showing the genes with highest loading scores. The score is indicated by y-axis and colour of circles. (**F**) Correlation plot of AR+ (green) and AR− (blue) samples using abundance values of 88 genes from the dataset GSE10167. In the correlation plot, the samples are the nodes labelled by sample ID and the edge thickness is proportional to the correlation score.

**Figure 4 biomolecules-14-00087-f004:**
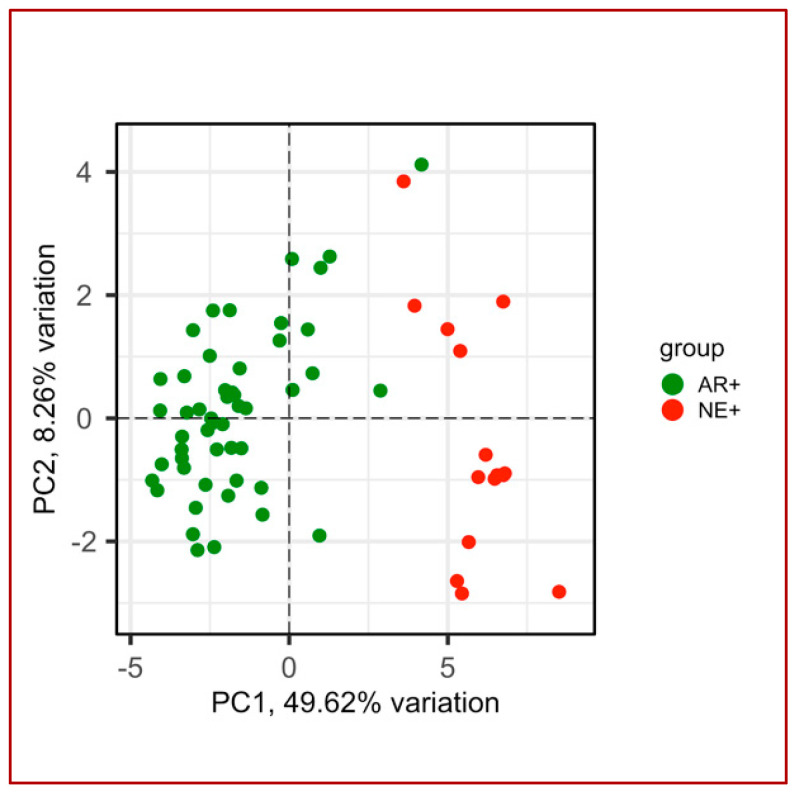
PCA plot of AR+ (green) and NE+ (red) samples using abundance values from the dataset GSE77930 of 30 genes forming the CRPC-gene set. PC1 and PC2 variation percentages are also shown.

**Figure 5 biomolecules-14-00087-f005:**
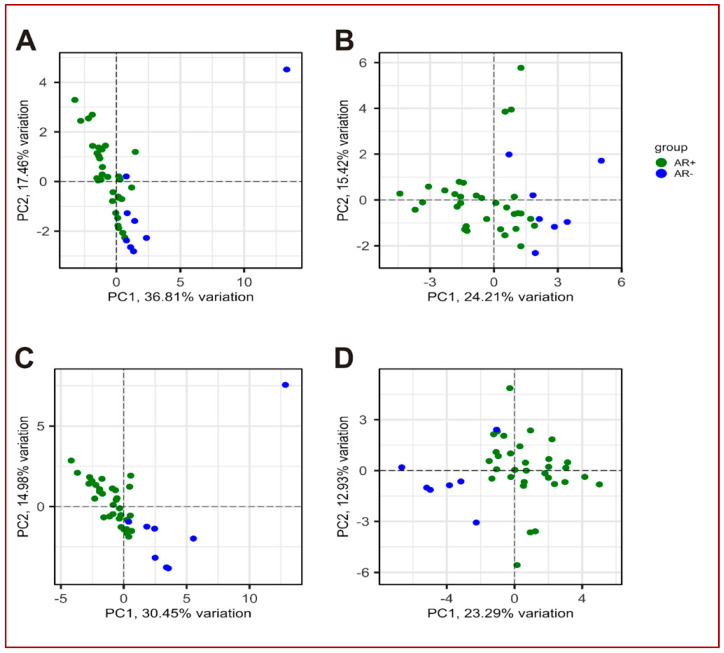
PCA plots showing the separation of AR+ (green) and AR− (blue) samples using expression values of 20 genes identified by [27] (**A**,**B**) and CRPC-gene set (**C**,**D**). As one of the samples was particularly distant from the others, it was removed in favour of the visualisation (**B**,**D**).

**Figure 6 biomolecules-14-00087-f006:**
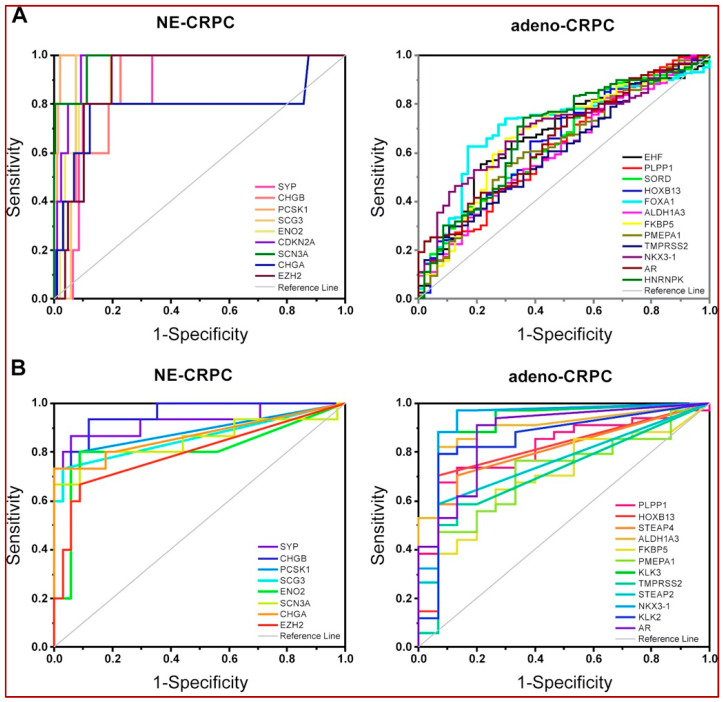
Validation of diagnostic efficiency of the CRPC-gene set. ROC curves were performed to classify between CRPC with neuroendocrine (NE) or adenoma (Adeno) phenotype using mRNA expression data from SU2C/PCF (208 patients) (**A**) and the Neuroendocrine-PC (49 patients) (**B**) studies. Only significant AUC values > 0.6 (asymptotic probability < 0.05) are reported.

**Figure 7 biomolecules-14-00087-f007:**
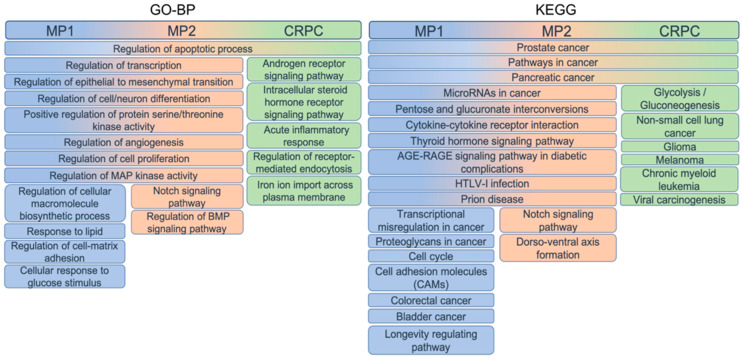
GO-BP (**left**) and KEGG (**right**) enrichment analysis. The top 50 enriched terms according to the number of overlapping genes were extracted. Redundant terms were solved by aggregating them into a single definition. The three gene sets are highlighted by different colours, which define the sharing of each single term along with the length of the box.

**Figure 8 biomolecules-14-00087-f008:**
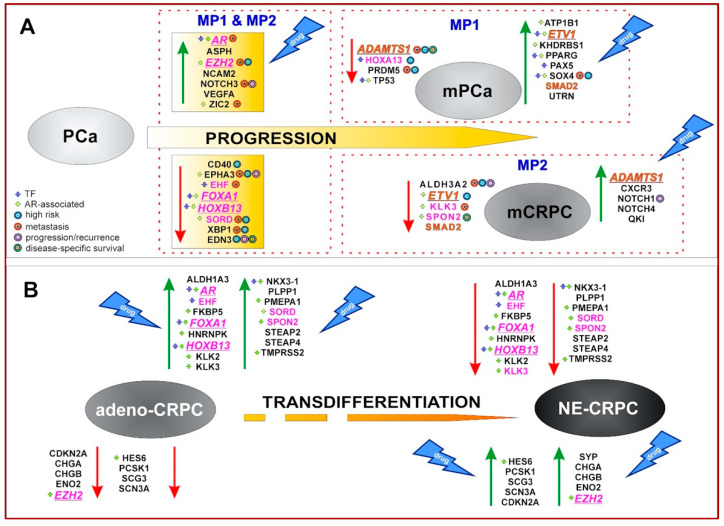
Schematic representation of the PCa- and CRPC-gene sets role in the different phases of prostate carcinogenesis, summarizing all the insights and results obtained in this work. (**A**) PCa-gene set involved in PCa progression to metastatic hormone-sensitive (mPCa) or castration-resistant (mCRPC) phenotypes. (**B**) Differential expression of CRPC-gene set during neuroendocrine transdifferentiation and their association with Adeno-CRPC (AR-dependent) and NE-CRPC (AR-independent) phenotypes. Green arrow: upregulation; red arrow: downregulation. Pink test: genes shared between PCa- and CRPC-gene sets. Orange test: PCa-genes with opposite expression in MP1 and MP2. Underlined and italic text: essential gene. TF, transcription factor.

## Data Availability

The publicly archived datasets used in this study are listed and referenced in Section 2.

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
