# Peer review of "Identification of Molecular Markers Associated with Prostate Cancer Subtypes: An Integrative Bioinformatics Approach"

_biomolecules, 2024, doi:10.3390/biom14010087_

Round 1
Reviewer 1 Report
Comments and Suggestions for Authors
Author Response
Comment 1: The objective of this research was to discover novel markers and drugs for castration-resistant prostate cancer patients or targeting metastasis. To achieve this goal, the authors conducted a computational analysis to find and evaluate the impact of two sets of genes associated with either cancer progression (PCagene set) or drug resistance (CRPC-gene set). The outcomes of their study offer potential biomarkers and therapeutic compounds for addressing the malignancy and resistance of prostate cancer. In general, this research presents a valuable and systematic computational approach to prostate cancer treatment, which has potential significance for clinical applications. However, many of the biomarkers or therapeutic compounds identified in this study have been previously reported, and there is a lack of actual experimental validation for any newly identified candidates discussed in this paper. Additionally, the primary and concluding outcome of this study consists of two gene lists associated with prostate cancer metastasis or resistance, each of them containing ~30 genes. While the author attempts to show their functions in Figure 8, the method for creating this figure is unclear. Is this figure solely based on gene expression data? Exploring possible interconnections among some of the representative genes could enhance the paper's value compared to just listing genes.
Response 1: We are very grateful to the reviewer for the comments that improved our manuscript. The idea that targeting more than one gene involved in cancer progression is more effective, was highlighted in recent works (doi.org/10.1080/17460441.2022.2072827; doi: 10.3389/fonc.2021.689131, doi: 10.1007/s12032-022-01905-7). The activation of bypass pathways and the complex nature of signalling networks that contain feedback loops and cross-talk are responsible for the poor effectiveness of monotherapy. The finding that therapy effectiveness and some observed drugs' contradictory effects are tumour-dependent is another issue to be considered in designing new therapeutic strategies. Consequently, the identification of clinically relevant predictive biomarkers is necessary to select the patients who may benefit from specific treatment. Taking into account these findings, we have attempted to achieve a systematical identification of different gene expression patterns associated with a significant difference in clinical outcome. In Figure 8, to clarify the role of the PCa and CRPC gene set in prostate carcinogenesis, we have collected all the insights and results obtained in this work (e.g. their changes in gene expression, drug targeting results) and schematically represented what we hypothesize about their involvement in the different phases of the disease. According to the reviewer’s comment, we added some more details in the Introduction (lines 110-115) and Discussion (lines 632-639) and a better explanation of the figure in its caption and in the Discussion (lines 526-529).
Regarding the interconnections, the enrichment analysis (Figure 7) made with the three gene sets (MP1, MP2 and CRPC) evaluates their co-involvement in biological processes and pathways. To provide the results of these functional interconnections, as the reviewer rightly suggested, we added the Supplementary_file_2.xlsx reporting for each gene-set the enrichment significative results (adjusted p-value ≤ 0.05) with GO-BP and KEGG terms and also specified this important aspect in paragraph 3.5 with the sentence “The functional role of the genes composing the three gene sets, as well as their co-involvement in processes and pathways, was assessed by performing the enrichment analysis of Gene Ontology Biological Processes (GO-BP) and KEGG pathways terms and selecting the top 50 in terms of overlapping genes” (lines 384-387).
Comment 2: In the ROC, AUC, and KL figures, the authors may want to show the number of samples, the cohort of patients, and the p-value/q value.
Response 2: In the legend of Figure 2 and Figure S1, and in the footnotes of Table S8 and Table S10 of the revised manuscript we have reported what was required.
Comment 3: In the last sentence on top of Figure 2, “… SPON2 and EDN3 was associated with high-risk PCa-specific mortality (Figure 1D)”, should be (Figure 2D)?
Response 3: Yes, it’s correct. In the revised manuscript we have corrected this error (line 271).
Comment 4: In Figure 3 A and F, it is hard to understand. It appears that both AR- and AR+ samples are interconnected with many others without clear differentiation or separation. 4. Are there any overlapped genes between Nelson list and CRPC-gene set?
Response 4: We thank the reviewer for the comment as it pointed out that the explanation of the figures was not clear enough. The figure 3 shows an important intermediate step that leads us to obtain the CRPC-gene set. In particular, the panels 3B (I think the reviewer meant 3B and not 3A) and 3F show samples correlation network performed by using the expression values of the entire transcriptome (3B) and the 88 genes obtained by adding to the 50-PCa gene set the lists reported in Robinson et al. (https://doi.org/10.1016/j.cell.2015.05.001) and Ylitalo et al. (https://doi.org/10.1016/j.eururo.2016.07.033) as shown in the Venn diagram (Figure 3C). The networks have the nodes representing the samples and the width and length of the edges representing the correlation scores. This means that the more the samples are correlated, based on the mRNA abundance values, the more they are close and thicker the edges. The 88 genes seem to better explain the belonging of samples to AR+ or AR- groups. Indeed, in the case of using the expression values of the 88 genes, the AR- samples are mainly in the left part of the network, while all the AR+ samples are distant in the right one (Figure 3F). Instead, in the case of the whole set of genes, the samples are all mixed up (Figure 3B). Furthermore, if we look at the discrimination power of these genes, we can see from the PCA plots (3A and 3D) that 88 genes show the same ability to discriminate the two groups of the whole transcriptome, suggesting that we rightly filtered and selected the genes. As we understand that it was not clearly explained we modified the text in the manuscript accordingly (lines 297-310).
Reviewer 2 Report
Comments and Suggestions for Authors
The authors performed integrative analysis of PCa gene expression data and evaluated genes involved into several key gene sets of PCas, such as PCa-gene sets and CRPC-gene sets. They further evaluated the drug repositioning possibilities for these genes showing differential expression among different subtypes of PCa. It seems that the currently study assumed that these genes in the candidate gene sets would be essential for the differentiation of different subtypes of PCas, which may not be true. This study is similar to a supervised study and only focused on previously published gene sets related to PCa. In fact, the more efficient way to analyze these data sets would be to normalize all PCa data sets and conduct unsupervised clustering analysis or other unbiased screening to differentiate subtypes of PCas.
(1) In the Materials and Methods section, these 8 gene expression data sets downloaded from the Gene Expression Omnibus (GEO) require to be explained with more details, such as what sequencing platforms were applied for generating these data and how these expression data from different flatforms were normalized before combination.
(2) It is also to mention the selection and exclusion of publicly available gene expression data for current study.
(3) It is necessary to use more stringent criterion, such as FDR, to filter genes for selecting genes for downstream integrative analysis. In the manuscript, the authors only used the relaxed fitlers of |log2fold-chang}>=1 and adjusted P value < 0.05. Did the adjusted P value refer to FDR generated by GEO2R? What kind of differential gene expression test was used to identify differentially expressed genes via GEO2R?
(4) In terms of SU2C/PCF dataset, the RNAseq data analysis should be performed according to the best practice of using read count with well recognized differential gene expression analysis software such as edgeR or others. The using of t-test is not suggested, since the more appropriate methods woud be applying negative binomial model to determine differentially expressed genes between two groups.
(5) The gene symbols, such as ADAMTS1, SPON2 and EDN3, need to be italicized.
Comments on the Quality of English Language
The manuscript was written well in English.
Author Response
Comment 1: The authors performed integrative analysis of PCa gene expression data and evaluated genes involved into several key gene sets of PCas, such as PCa-gene sets and CRPC-gene sets. They further evaluated the drug repositioning possibilities for these genes showing differential expression among different subtypes of PCa. It seems that the currently study assumed that these genes in the candidate gene sets would be essential for the differentiation of different subtypes of PCas, which may not be true. This study is similar to a supervised study and only focused on previously published gene sets related to PCa. In fact, the more efficient way to analyze these data sets would be to normalize all PCa data sets and conduct unsupervised clustering analysis or other unbiased screening to differentiate subtypes of PCas.
Response 1: We thank the reviewer for the comments. Regarding the essentiality analysis, the knock-out screening experiments on human genes are performed on tumoral cell lines in vitro. We are aware that what is observable in in vitro cultures could not be exactly reproduced in an alive organism and that these kinds of results are an approximation. Nonetheless, in vitro experiments are still considered an important and valuable strategy to get useful insights and the variety of the genetic background of the different cell lines gives the possibility to evaluate the biological behaviour in different scenarios. In our humble opinion, we think that what we observed through the essentiality analysis is an additional clue for the contribution of the selected genes in discriminating the considered cancer groups.
Regarding the second point, the aim of our work was not to stratify prostate cancer patients but rather to identify candidate biomarkers for pre-defined clinically relevant phenotype classes (PCa, mPCA, Adeno-CRPC and NE-CRPC). To this extent, we adopted a consensus strategy among publicly available independent datasets. As we pointed out in the Introduction, the integration of independent datasets from different studies and experiments at the level of raw data requires not a simple normalization but a batch effect correction, that, especially in the case of batches representing different methods, platforms, laboratories and personnel, means removing noise with the risk of also removing biological signals. For this reason, the application of batch correction is still an open discussion for the scientific community (https://doi.org/10.1016/j.tibtech.2017.02.012; https://doi.org/10.1016/j.jgg.2019.08.002). Considering the limits of batch correction and the aim of our work, we integrated the study not at raw data but at the results level of differential expression analysis by searching for a consensus among all. This approach allowed us to consider both microarray and RNA sequencing data. Furthermore, the gene sets identified have been used to perform PCA analysis, which is unsupervised, and show their discrimination power in separating the patient groups. As we considered the reviewer’s comments a useful suggestion for improving the description of our aim and strategy, we made some changes in the text of the Introduction section (lines 88-90, 106-108).
Comment 2: In the Materials and Methods section, these 8 gene expression data sets downloaded from the Gene Expression Omnibus (GEO) require to be explained with more details, such as what sequencing platforms were applied for generating these data and how these expression data from different flatforms were normalized before combination.
Response 2: We thank the reviewer for the suggestion. We added the codes of the platforms accordingly, as some datasets have more than one. For more details on the datasets, we added the link to the repository where the data and methods are described (Section 2.1). Regarding the normalization, as described in the manuscript, we did not integrate the datasets at the data level but we searched for a consensus among the results of differential expression analysis, since for the first case a batch effect correction is needed and this is an open issue as it involves several risks of data alteration as mentioned and referenced previously.
Comment 3: It is also to mention the selection and exclusion of publicly available gene expression data for current study.
Response 3: The choice of the datasets used in this study was guided by the availability of data from the patient's cancer profiles of interest: PCa, mPCa, and CRPC with or without neuroendocrine (NE) features.
Comment 4: It is necessary to use more stringent criterion, such as FDR, to filter genes for selecting genes for downstream integrative analysis. In the manuscript, the authors only used the relaxed fitlers of |log2fold-chang}>=1 and adjusted P value < 0.05. Did the adjusted P value refer to FDR generated by GEO2R? What kind of differential gene expression test was used to identify differentially expressed genes via GEO2R?
Response 4: The adjusted p-value refers to Benjamini & Hochberg (False discovery rate) adjustment for multiple testing. We added the detail of the type of adjustment where needed.
GEO2R uses Limma R package to analyze differentially expressed genes from microarray data. Limma is a stable and widely applicable method based on use of linear models to assess differential expression in the context of multi-factor designed experiments. We added this detail in Section 2.2.
Comment 5: In terms of SU2C/PCF dataset, the RNAseq data analysis should be performed according to the best practice of using read count with well recognized differential gene expression analysis software such as edgeR or others. The using of t-test is not suggested, since the more appropriate methods woud be applying negative binomial model to determine differentially expressed genes between two groups.
Response 5: We agree with the reviewer about the best practice for comparing groups and analyzing the differentially expressed genes. Unfortunately, the SU2C/PCF dataset does not contain raw read count data, only RPKM, standardized or not by Z-score. The widely used software, such as edgeR, DeSeq2 or Limma are essentially thought and developed to work with no pre-normalized data, as they apply their normalization strategy. Even when it is possible to skip the normalization step, as it is for edgeR and Limma, it is not recommended to use them with RPKM data. The variance in the negative binomial model depends on the mean, so the significance of any differences will depend on the absolute size of the count. This information about the absolute size is lost upon normalization. To convert RPKM to read counts we should know how the RPKM have been calculated as they involve the transcript length, but this information is not available. We are aware that the t-test is not the election choice, but it is also the strategy applied by cBioPortal in the “Comparison Groups” tool, that we used now to be totally in accordance with what is reported in the repository of the dataset. It is also worth mentioning that the t-test results are used in the consensus strategy, this means that the results must be confirmed by at least one more dataset and analysis to be taken into account. By using the cBioPortal tool, we adjusted the p-value of the t-test by FDR method (q-value) and considered only the results with q-value < 0.05. We confirmed all the genes identified before and added a new one, SYP, thus that the CRPC gene set now contains 30 genes. We added the gene name in the text (line 341) and the results in tables and figures accordingly.
Modified tables and figures: Figure 6, Figure 8, Figure S1, Figure S4, Table S9, Table S10.
Comment 6: The gene symbols, such as ADAMTS1, SPON2 and EDN3, need to be italicized.
Response 6: We thank the reviewer for the observation and italicized the gene names throughout the manuscript.
Comments on the Quality of English Language: The manuscript was written well in English.
Thanks for the comment
Reviewer 3 Report
Comments and Suggestions for Authors
In this article, by exploring multiple datasets and an integrative bioinformatics approach, the authors found a series of key genes involved in PCa progression and CRPC evolution. Meanwhile, the genes identified were able to distinguish the different subtypes of CRPC. This article is ready to be published before completing the following revisions.
1. With regard to “CHGA positive/SYP positive/SR negative groups” in “2.2 Differential expression analysis” section, is "SR" a misspelling of "AR"?
2. GSEA analysis is written in the method “2.3 Over-representation analysis”, however GSEA is not part of the over-representation analysis. Furthermore, the authors should have been more specific in writing how 32 genes can be used to obtain 43 genes through GSEA.
3. The data used throughout the paper are from bulk RNA-seq data, but due to the heterogeneity of gene expression in prostate cancer cells, the authors could verify these genes with single-cell sequencing.
4. Although several gene sets were obtained in the article, the authors used the individual genes contained in gene sets to predict prognosis and distinguish subtypes. Because of the unreliability of single gene, the authors should integrate these genes together into a signature, which can be used to distinguish between different CRPC phenotypes and predict prognosis better.
Author Response
In this article, by exploring multiple datasets and an integrative bioinformatics approach, the authors found a series of key genes involved in PCa progression and CRPC evolution. Meanwhile, the genes identified were able to distinguish the different subtypes of CRPC. This article is ready to be published before completing the following revisions.
We thank the Reviewer for taking the time to review this manuscript.
Comment 1: With regard to “CHGA positive/SYP positive/SR negative groups” in “2.2 Differential expression analysis” section, is "SR" a misspelling of "AR".
Response 1: Yes, it is. We thank the reviewer for noticing it and corrected the gene name accordingly.
Comment 2: GSEA analysis is written in the method “2.3 Over-representation analysis”, however GSEA is not part of the over-representation analysis. Furthermore, the authors should have been more specific in writing how 32 genes can be used to obtain 43 genes through GSEA.
Response 2: We know that often the over-representation and enrichment analyses are considered two different approaches due to the different statistics underlying them and the fact that the first considers only the differentially expressed genes, but they are substantially the same thing. Both analyze the set of genes that overlap or are over-represented in large sets and may have an association with different phenotypes. We computed the overlap (synonym of over-representation) between our set of genes and those collected in MSigDB. The term “over-represented” is used both on the GSEA home page “The GSEA algorithm simply asks if the genes in a gene set are overrepresented at the top or bottom of the ranked list of genes from the expression dataset.” and in the relative paper to explain the calculation of the Enrichment Score (https://doi.org/10.1073/pnas.0506580102).
In agreement with the second comment, we provided more details on GSEA analysis in the “2.3 Over-representation analysis” (lines 174-182).
Comment 3: The data used throughout the paper are from bulk RNA-seq data, but due to the heterogeneity of gene expression in prostate cancer cells, the authors could verify these genes with single-cell sequencing.
Response 3: We thank the reviewer for the comment that opened an interesting discussion and suggested adding some important considerations into the main text. Single-cell sequencing is a very important tool for identifying sub-populations by discovering unique characteristics of individual cells. While we recognize its great potential we think that our work relies on different assumptions and different aims. We did not pursue the stratification of samples and we did not claim to reveal new sub-populations, rather we used pre-defined and clinically relevant phenotypes (PCa, mPCA, Adeno-CRPC and NE-CRPC) to search for a consensus of altered genes among independent datasets (both from microarrays and RNA-seq data). In front of a recognized heterogeneity and plasticity of cell populations in PCa patients, we think that it is important to find targets that globally characterize the phenotypes that are currently classified in clinical practices.
Our scientific questions were:
- what are the set of genes that in concert are altered in the four cancer sub-types and contribute to their progression?
- are these genes effective biomarkers targettable by drugs already used in oncology practices?
Differential expression among cell populations would certainly reveal interesting insights about the characterization of the individual cell groups but would not reply to these questions.
Nonetheless, we agree that it would be interesting to investigate the behaviour of our gene sets in cell subpopulations but this would be an intensive additional work moved from a different aim and for this reason and for not weighing the reading of the manuscript down, we think to include it in future work.
Comment 4: Although several gene sets were obtained in the article, the authors used the individual genes contained in gene sets to predict prognosis and distinguish subtypes. Because of the unreliability of single gene, the authors should integrate these genes together into a signature, which can be used to distinguish between different CRPC phenotypes and predict prognosis better.
Response 4: We thank the reviewer for the useful suggestion. We added the Kaplan-Meier plot made with the set of the three genes used previously (please see new Fig. 2D) and described the method, accordingly (lines 204-211).
Reviewer 4 Report
Comments and Suggestions for Authors
Well written paper analysing transcriptomic changes in prostate cancer samples. Many genes identified as involved have been described before.
Specific points:
Page 12, 3.6: It is surprising that AR is not essential for VCaP, please comment
Page 13, line 3: It should be mentioned that 17-AAG is an HSP90 inhibitor. Its impact on EZH2, PAX5 and KHDRBS1 should be better explained.
Page 13, 4th paragraph and Table S11: How was this list selected? It contains many cytotoxic agents even though the scope of the study presented is to identify novel, tumor-specific targets. Also, three HDAC inhibitors are mentioned but this target does not appear in the presented study. There is a brief mention in the Discussion, but the impact on AR and downstream gene expression is certainly very general. Same question for BRAF, ERBB, MEK: none of these targets came out of the presented integrative approach, so why are they discussed? Conversely, EZH2 is highligthed as upregulated in NE-CRPC and also included in the Discussion, but the approved inhibitor tazemostat is not mentioned or discussed even though there are ongoing clinical studies in prostate cancer.
Page 13, 3rd paragraph: what is meant by “genes regulating AR activity”? I am not aware that TMPRSS2, ALDH1A3 or PMEPA1 impact AR activity.
Page 13, 4th paragraph: FOXA1 and HOXB13 are pioneer factors found in chromatin remodeler complexes but they do not encode them.
Page 17, 4., last paragraph: it says that panobinostat, vorinostat, dasatinib and temsirolimus are included in Table S9, which is wrong. They are found in S11, but as mentioned above the corresponding targets (HDACs) did not come out of the presented study.
General: recent important studies on molecular profiling of prostate cancer are not discussed and not even referenced, for example MP Labrecque et al, J Clin Invest, 2019 or N Menssouri et al, Clin Cancer Res, 2023.
Comments on the Quality of English LanguagePage 2, line 3 and page 5, fig. 1: insurgence à emergence
Page 12, 3.6 and page 13, line 13: ADMATS1 à ADAMTS1
Author Response
Well written paper analysing transcriptomic changes in prostate cancer samples. Many genes identified as involved have been described before.
We thank the reviewer for taking the time to review our manuscript.
Comment 1: Page 12, 3.6: It is surprising that AR is not essential for VCaP, please comment
Response 1: VCaP cell line highly expressed AR, as also confirmed in our results (Please see Fig.S2 third column). Nonetheless, it has been reported that VCaP cells do not require androgens for in vitro growth although they are sensitive to the presence of androgens (The Prostate 57:205 (2003) DOI 10.1002/pros.10290). AR signalling is not necessary for the proliferation of VCaP cells and it has been demonstrated that treatment with an AR antagonist did not affect the proliferation of VCaP cells (https://doi.org/10.4196/kjpp.2015.19.3.235). Furthermore, VCaP cells express both wild-type full-length AR and AR-V7 splice variants and are included in the androgen-independent/androgen-responsive group of in vitro models of CRPC. This independence may be related to the alteration of TP53 and RB genes and the TMPRSS2–ERG fusion gene observed in VCaP cells that could globally activate different pathways, other than that of AR, to control cell growth. In line with this hypothesis, it has been recently found that VCaP cells coexpress AR and neuroendocrine genes (Clin Cancer Res; 29(21), 2023. DOI: 10.1158/1078-0432.CCR-22-3736). A comment on this topic was included in the revised manuscript (lines 432-434).
Comment 2: Page 13, line 3: It should be mentioned that 17-AAG is an HSP90 inhibitor. Its impact on EZH2, PAX5 and KHDRBS1 should be better explained.
Response 2: please see below Response to 2) and 3)
Comment 3: Page 13, 4th paragraph and Table S11: How was this list selected? It contains many cytotoxic agents even though the scope of the study presented is to identify novel, tumor-specific targets. Also, three HDAC inhibitors are mentioned but this target does not appear in the presented study. There is a brief mention in the Discussion, but the impact on AR and downstream gene expression is certainly very general. Same question for BRAF, ERBB, MEK: none of these targets came out of the presented integrative approach, so why are they discussed? Conversely, EZH2 is highligthed as upregulated in NE-CRPC and also included in the Discussion, but the approved inhibitor tazemostat is not mentioned or discussed even though there are ongoing clinical studies in prostate cancer.
Response to 2) and 3): As both the reviewer's comments 2) and 3) pointed out that the drug identification section has not been adequately explained in our manuscript, we decided to merge the answers to both as they concern the same subject. Our study attempts to identify the set of genes specifically associated with clinically relevant phenotypes of PCa to facilitate the development of more effective therapies, especially for advanced stages. The idea that targeting more than one gene involved in cancer progression is more effective, was highlighted in recent works (doi.org/10.1080/17460441.2022.2072827; doi: 10.3389/fonc.2021.689131, doi: 10.1007/s12032-022-01905-7). As reported in the Introduction (page 2, 2th paragraph), using an alternative approach, we identify not a single marker but a group of genes associated with different PCa phenotypes (PCa gene set, CRPC gene set). In this perspective, our study aimed also to select, among the compounds already in clinical use for other diseases, those targeting multiple proteins. For this purpose, using the GSCA platform and evaluating only the overexpressed genes of PCa or CRPC gene sets, we identified the drugs with the highest correlation between gene expression and drug sensitivity (Figure S3, S4 and S5). Then we selected and reported in Table S11 the drugs approved by the FDA among those identified with GSEA to highlight the compounds already in use in oncology that could also be considered for PCa therapy. Taking into account the reviewer's comments, we performed additional analysis to clarify the results reported in Table S11. Using the STRING database, the interaction between the up-regulated genes in PCa or CRPC AR+ or CRPC AR- (Table S8, Table S10), and the drug targets reported in Table S11 was evaluated and reported in Figure S6 of the revised manuscript. Overall, multiple connections between the gene sets and the drug targets are supported by a very low enrichment p-value, suggesting a strong functional relation among them. From this observation we can assume that compounds targeting a single protein, which in turn interacts with different proteins, can have effects on different other pathways (DOI: 10.1002/psp4.12861). Similar results were obtained analyzing proteins coded by up-regulated PCa gene set and HSP90, target of 17-AAG (figure S7)
Globally, our results are in agreement with the hypothesis that a single drug targeting multiple pathways can improve therapeutic efficiency. The activation of bypass pathways and the complex nature of signalling networks that contain feedback loops and cross-talk are responsible for the poor effectiveness of monotherapy. The assumption that therapy effectiveness and some observed drugs' contradictory effects are tumor-dependent is another issue to be considered in designing new therapeutic strategies. Consequently, the identification of clinically relevant predictive biomarkers is necessary to select the patients who may benefit from specific treatments.
Finally, GSCA’S drug datasets do not contain tazemostat, so it is not included in our analysis.
From the above we have revised the manuscript adding some more details in the Introduction (lines 110-115), in the 3.7 Computational drug identification based on the PCa- and CRPC-gene sets section (lines 469-483), in the Discussion (lines 632-639 and 668-673)) and adding two supplementary figures (Figure S6 and S7).
Comment 4: Page 13, 3rd paragraph: what is meant by “genes regulating AR activity”? I am not aware that TMPRSS2, ALDH1A3 or PMEPA1 impact AR activity.
Response 4: In agreement with the Reviewer’s comment the sentence in the revised manuscript has been substituted by “genes associated with AR activity” (line 572).
Comment 5: Page 13, 4th paragraph: FOXA1 and HOXB13 are pioneer factors found in chromatin remodeler complexes but they do not encode them.
Response 5: We thank the Reviewer for the correction, in the revised text we have rephrased it as follows “FOXA1 and HOXB13, acting as pioneer factors (PFs) found in chromatin remodeler complexes, reprogram AR transcriptional activity.” (line 593).
Comment 6: Page 17, 4., last paragraph: it says that panobinostat, vorinostat, dasatinib and temsirolimus are included in Table S9, which is wrong. They are found in S11, but as mentioned above the corresponding targets (HDACs) did not come out of the presented study.
Response 6: In the revised manuscript we have corrected the error (line 687).
Comment 7: General: recent important studies on molecular profiling of prostate cancer are not discussed and not even referenced, for example MP Labrecque et al, J Clin Invest, 2019 or N Menssouri et al, Clin Cancer Res, 2023.
Response 7: Agreeing with the Reviewer’s suggestion we added the references in the revised manuscript:
ref n°73 MP Labrecque et al, J Clin Invest https://doi.org/10.1172/JCI128212.
ref n°74 N Menssouri et al, Clin Cancer Res, 2023. doi: 10.1158/1078-0432.CCR-22-3736
Comments on the Quality of English Language:
1) Page 2, line 3 and page 5, fig. 1: insurgence à emergence
R1: We thank the reviewer for the correction and changed the text and the figure accordingly.
2) Page 12, 3.6 and page 13, line 13: ADMATS1 à ADAMTS1
R2: We thank the reviewer and corrected the gene name.
Round 2
Reviewer 1 Report
Comments and Suggestions for Authors
The authors have addressed my questions properly in their revised submission.
Author Response
Comment: The authors have addressed my questions properly in their revised submission.
Response: We thank the reviewer for the positive feedback and for the time dedicated to the revision of our work
Reviewer 2 Report
Comments and Suggestions for Authors
I think the authors have tried their best to improve the manuscript. Please add more limitations based on your responses to my previous questions to the limitation section. Once completed the limitation section, it would is suitable for publication.
Author Response
Comment 1: I think the authors have tried their best to improve the manuscript. Please add more limitations based on your responses to my previous questions to the limitation section. Once completed the limitation section, it would is suitable for publication.
Response 1: We thank the reviewer for his comment and the additional useful suggestion. We added a sentence about the essentiality analysis in the Discussion (lines 679-685), and a general comment regarding the limitations of the current study in the Conclusions section (lines 703-709).
Reviewer 4 Report
Comments and Suggestions for Authors
The authors have answered my points very carefully. Very nice data.
Author Response
Comment: The authors have answered my points very carefully. Very nice data.
Response: We thank the reviewer for the comment and the effort put into the revision of our work.